# Enhancing Erythropoiesis by a Phytoestrogen Diarylheptanoid from *Curcuma comosa*

**DOI:** 10.3390/biomedicines10061427

**Published:** 2022-06-16

**Authors:** Kanit Bhukhai, Guillemette Fouquet, Yutthana Rittavee, Nopmullee Tanhuad, Chaiyaporn Lakmuang, Suparerk Borwornpinyo, Usanarat Anurathapan, Apichart Suksamrarn, Pawinee Piyachaturawat, Arthit Chairoungdua, Olivier Hermine, Suradej Hongeng

**Affiliations:** 1Department of Physiology, Faculty of Science, Mahidol University, Bangkok 10400, Thailand; tnopmullee@gmail.com (N.T.); pawinee.pia@mahidol.ac.th (P.P.); arthit.chi@mahidol.ac.th (A.C.); 2Institut Hospitalo-Universitaire Imagine, Université Sorbonne Paris Cité, Assistance Publique-Hôpitaux de Paris, Hôpital Necker, 75015 Paris, France; guillemette.fouquet@gmail.com (G.F.); ohermine@gmail.com (O.H.); 3INSERM U1163 and CNRS ERL8254, Université Paris Descartes, Faculté de Médecine, Hôpital Necker, 75015 Paris, France; 4Department of Biology, Faculty of Science, Mahidol University, Bangkok 10400, Thailand; rittavee.y@gmail.com; 5Department of Chemistry, Faculty of Science, Mahidol University, Bangkok 10400, Thailand; chaiyaporn.lakmuang@gmail.com; 6Department of Biotechnology, Faculty of Science, Mahidol University, Bangkok 10400, Thailand; bsuparerk@gmail.com; 7Excellent Center for Drug Discovery, Mahidol University, Bangkok 10400, Thailand; 8Department of Pediatrics, Faculty of Medicine, Ramathibodi Hospital, Mahidol University, Bangkok 10400, Thailand; usanarat.anu@mahidol.ac.th; 9Department of Chemistry and Center of Excellence for Innovation in Chemistry, Faculty of Science, Ramkhamhaeng University, Bangkok 10240, Thailand; s_apichart@ru.ac.th; 10Laboratory of Excellence GReX, 75015 Paris, France; 11Service d’Hématologie Clinique Adultes, Assistance Publique-Hôpitaux de Paris, Hôpital Necker, 75015 Paris, France

**Keywords:** anemia, erythropoiesis, erythropoietin receptor signaling, diarylheptanoid, phytoestrogen

## Abstract

Erythropoietin (Epo) is widely used for the treatment of anemia; however, non-hematopoietic effects and cancer risk limit its clinical applications. Therefore, alternative molecules to improve erythropoiesis in anemia patients are urgently needed. Here, we investigated the potential effects of a phytoestrogen diarylheptanoid (3*R*)-1,7-diphenyl-(4*E*,6*E*)-4,6-heptadien-3-ol, (ASPP 049) isolated from *Curcuma comosa* on promoting erythropoiesis. Treatment with *C. comosa* extract improved anemia symptoms demonstrated by increasing red blood cell numbers, hematocrit, and hemoglobin content in anemic mice. In addition, ASPP 049, the major compound isolated from *C. comosa*, enhanced the suboptimal Epo dosages to improve erythroid cell differentiation from hematopoietic stem cells, which was inhibited by the estrogen receptor (ER) antagonist, ICI 182,780. Moreover, the ASPP 049-activated Epo-Epo receptor (EpoR) complex subsequently induced phosphorylation of EpoR-mediated erythropoiesis pathways: STAT5, MAPK/ERK, and PI3K/AKT in Epo-sensitive UT-7 cells. Taken together, these results suggest that *C. comosa* extract and ASPP 049 increased erythropoiesis through ER- and EpoR-mediated signaling cascades. Our findings provide insight into the specific interaction between a phytoestrogen diarylheptanoid and Epo-EpoR in a hematopoietic system for the potential treatment of anemia.

## 1. Introduction

Anemia is the most common blood disorder, affecting approximately 32.9% of the world’s population [1]. It is characterized by insufficient numbers of red blood cells (RBCs) and low hemoglobin concentrations and causes insufficient delivery of oxygen to meet an individual’s physiological needs. The pathophysiology of anemia is diverse and often results from multifactorial causes, leading to an imbalance in erythrocyte loss relative to production, which can be due to ineffective erythropoiesis [2,3]. Erythropoietin (Epo), a key growth factor for RBC production (erythropoiesis), is frequently used for the treatment of anemia. Binding of Epo to the Epo receptor (EpoR) activates three major signaling pathways involved in erythropoietic differentiation: signal transducer and activator of transcription 5 (STAT5), mitogen-activated protein kinase (MAPK)/extracellular signal-regulated kinase (ERK), and phosphoinositide-3-kinase (PI3K)/protein kinase B (AKT) [4]. Several recombinants and analogs of Epo have become widely used for treatment of anemia patients [5,6]. However, Epo treatment is ineffective in some cases, has non-hematopoietic effects, and carries risks when administered at high dosages [7,8]. Therefore, identification of more effective drug-targeting pathways involved in erythropoiesis is needed. Estrogen (E2), a female sex steroid hormone, has been shown to induce proliferation and differentiation of hematopoietic stem and progenitor cells [9]. E2 improves hematopoietic differentiation of human-induced pluripotent stem cells by activation of E2-ERα signaling-enriched production of hemangioblasts, hematopoietic progenitors, and, subsequently, erythrocytes [9,10]. These similar effects were also observed in hematopoietic stem cells (HSCs) derived from human umbilical cord blood and were abolished by the ER antagonist ICI 182,780 [11]. In addition, a recent study revealed that mice treated with tamoxifen (an ER antagonist) exhibited decreased numbers of RBCs, Hb content, and hematocrit [12]. Although E2 effectively alleviates anemia, its potential to induce adverse effects limits its use. *Angelica sinensis*, a Chinese medicinal plant used to promote hematopoietic function in menopausal women [13], increases Epo mRNA expression and secretion to induce hematopoiesis through hypoxia-mediated hypoxia-inducible factor 1 α [14], MAPK, and ERα signaling [15,16]. Although the ability of phytoestrogens to promote HSC proliferation and/or differentiation has been demonstrated, roles of phytoestrogen on ER and erythropoiesis-related pathways remain unclear.

Phytoestrogens are non-steroidal compounds naturally occurring in plants that have estrogenic-like activities in many target tissues. (3*R*)-1,7-diphenyl-(4*E*,6*E*)-4,6-heptadien-3-ol (ASPP 049), a diarylheptanoid isolated from *Curcuma comosa* (*C. comosa*, Zingiberaceae family), reportedly possesses high estrogenic activities. ASPP 049 mediates its action via estrogen-responsive genes through the activation function 2 domain (AF2) of ERα and is abolished by ICI 182,780 [17]. In addition, *C. comosa* extract and ASPP 049 possess several pharmacological activities, including induction of osteoblastic cell differentiation through ER/Akt/GSK3-β-dependent activation of Wnt/β-catenin [18] and MAPK/ERK signaling pathways [19]. However, the effect of this compound in relation to erythropoiesis has not yet been investigated. The present study investigated the potential of *C. comosa* extract and its primary component, the diarylheptanoid ASPP 049, to promote erythropoiesis. Our results reveal a novel action of ASPP 049 that increased the sensitivity of erythroid cells to Epo through ERα and EpoR signaling by modulating activation of three major pathways involved in erythroid cell production: STAT5, MAPK/ERK, and PI3K/AKT.

## 2. Materials and Methods

### 2.1. Isolation of ASPP 049 from C. comosa

Plant extraction, compound isolation, and identification process were conducted as previously described [20]. Briefly, rhizomes of *C. comosa* were purchased from Kampangsan district, Nakhon Pathom province, Thailand. The rhizomes of *C. comosa* were taxonomically identified with a voucher herbarium specimen (SCMU 300) [21]. Then, they were sliced, dried, ground, and extracted with n-hexane. After solvent removal, diarylheptanoids were isolated from the hexane extract by column chromatography. Nonphenolic diarylheptanoid (ASPP 049), (3*R*)-1,7-diphenyl-(4*E*,6*E*)-4,6-heptadien-3-ol (Figure 2A), was isolated. The structure of ASPP 049 was confirmed by nuclear magnetic resonance and mass spectroscopy. The purity of ASPP 049 was estimated to be >98% by HPLC.

### 2.2. Anemia Mouse Model and Human HSCs

C57BL/6J mice were purchased from Envigo. The animal experimental protocol was approved by the French Institutional Committee, Institut Imagine, Université Sorbonne Paris Cité, Hôpital Necker, Paris, France (A75-15-34). All animal experiments were performed in accordance with relevant guidelines and regulations. Human CD34^+^ HSCs were obtained from human cord blood according to the Helsinki Declaration and with approval from the Ethics Committee of Institut Imagine, Université Sorbonne Paris Cité, Hôpital Necker (Comité de Protection des Personnes “Ile-de-France II”), Paris, France and the Ethical Clearance Committee on Human Rights Related to Research Involving Human Subjects at Ramathibodi Hospital, Mahidol University, Bangkok, Thailand (MURA2019/916). Informed consent was obtained from all subjects before performing the experiments.

### 2.3. C. Comosa Treatment of Anemic Mice

C57BL/6J mice were employed for induction of anemia. They were sub-lethally irradiated at 5 cGy using a cobalt-60 gamma source to deplete the bone marrow cells. Subsequently, the production of all types of blood cells and their related components including RBCs, hematocrit, hemoglobin, and reticulocytes were disrupted. Irradiated mice were randomly divided into three groups of nine animals each, then intraperitoneally injected with vehicle control or *C. comosa* extract (50 or 100 mg/kg of body weight (BW)) each day throughout the course of the experiment. Blood samples were collected from the retro-orbital plexus, and hematological parameters were determined with an MS9 analyzer (Melet Schloesing) at various time points.

### 2.4. Antibodies and Reagents

The following antibodies were used: anti-Epo receptor (kindly provided by Dr. Patrick Mayeux, Institut Cochin, Paris, France), anti-p44/42 ERK (phospho-ERK1 at Thr202 and Tyr204, and phospho-ERK2 at Thr185 and Tyr187), anti-ERK, anti-phospho-AKT (Ser473), anti-AKT, anti-phospho-STAT5 (Tyr694/699), anti-STAT5 (all from Cell Signaling Technology, Danvers, MA, USA), anti-β-actin (Merck, Darmstadt, Germany), and horseradish peroxidase (HRP)-conjugated secondary antibody (Santa Cruz Biotechnology, Dallas, TX, USA). The following reagents were used: 17β-estradiol (E2, Merck, Germany), ICI 182,780 (Tocris Bioscience, Bristol, UK), Complete Mini EDTA-free (Roche, Basel, Switzerland), and Immobilon Crescendo Western HRP substrate (Merck, Germany).

### 2.5. In Vitro Erythroblast Culture

CD34^+^ HSCs isolated from human cord blood and leftover specimens (Miltenyi CD34 Progenitor Cell Isolation Kit, Miltenyi Biotech, Bergisch Gladbach, Germany) were cultured in the presence of interleukin-3 (10 ng/mL), stem cell factor (100 ng/mL) (all from Peprotech, Rocky Hill, NJ, USA), and Epo (0.1 or 1 U/mL, obtained from the Department of Hematology, Hôpital Necker, Paris, France) in Iscove’s Modified Dulbecco’s Medium (Gibco, USA) supplemented with 15% BIT 9500 (Stem Cell Technologies, Vancouver, BC, Canada) as previously described [22]. For human colony-forming cell (CFC) assays, CD34^+^ cells were cultured in MethoCult^TM^ SF H4236 (Stem Cell Technologies, Canada) supplemented with interleukin-3 (10 ng/mL) and stem cell factor (50 ng/mL), as previously described [23], along with various concentrations of Epo, E2, and ASPP 049. Numbers of burst-forming unit-erythroid cells (BFU-Es) were scored after 14 days of culture according to the manufacturer’s recommendations (Stem Cell Technologies, Canada). For the colony-forming cell (CFC) assay, the BFU-Es were recognized from the colonies with the characteristic of hemoglobinized cells, which are the red or brown colonies containing more than 200 erythroblasts in single or multiple clusters.

Murine HSCs were negatively sorted from the bone marrow of C57BL/6J mice using a magnetic lineage cell-depletion kit from Miltenyi Biotec. Sorted murine lineage-negative (Lin^−^) cells were used for CFC assays employing MethoCult^TM^ SF M3236 and liquid culture after 7 days of treatment with indicated concentrations of E2 and ASPP 049. Numbers of BFU-Es were analyzed according to the manufacturer’s recommendations (Stem Cell Technologies, Canada) and cell viability was assessed by trypan blue exclusion assay.

### 2.6. Flow Cytometry Analysis

Antibodies used for cytometry were directed against human CD34 (clone 581, Biolegend, San Diego, CA, USA), human CD36 (clone 5-271, Biolegend, USA), human CD71 (clone REA902, Miltenyi Biotech, Germany), and human CD235a (glycophorin A, Miltenyi Biotec, Germany). Non-viable cells were excluded by 7AAD (Thermo Fisher Scientific, Waltham, MA, USA). Data were acquired with a FACS Canto Flow Cytometer instrument (BD Biosciences, East Rutherford, NJ, USA) and analyzed with FlowJo software (Tree Star).

### 2.7. Cytospin Preparations and Histological Staining

In vitro cultured erythroid cells were centrifuged onto slides for 3 min at 500 rpm using a Cytospin 4 (Thermo Scientific, USA), air dried, and fixed with methanol for 10 min. Next, cells were stained with Liu’s stain according to the manufacturer’s recommendations (BaSO Biotech, New Taipei City, Taiwan). Morphological characteristics of nuclei, enucleated RBCs, and mature RBCs were evaluated using light microscopy according to the manufacturer’s recommendations.

### 2.8. Erythroid Cell Starvation, Stimulation, and Western Blot Analysis

Epo-sensitive UT-7 (UT7/Epo) cells were maintained in Minimum Essential Medium Alpha (Gibco, Grand Island, NY, USA) supplemented with 10% fetal bovine serum (Hyclone, Logan, UT, USA) and 1 U/mL Epo at 37 °C in a 5% CO_2_ humidified atmosphere. To avoid exposure to growth factors, serum-starved cells were obtained by washing and culturing cells in a medium without fetal bovine serum for 16–18 h before experiments. Serum-starved UT-7/Epo cells were plated at a density of 2 × 10^6^ cells/mL and treated with varying concentrations of Epo, ASPP 049, or Epo combined with ASPP 049. At various time points, treated cells were washed twice with cold phosphate-buffered saline and then lysed with modified radioimmunoprecipitation assay buffer (in mM: 50 Tris-HCl (pH 7.4), 150 NaCl, 1 EDTA, 1 NaF, 1 Na_3_VO_4_, 1 PMSF; plus 1% TritonX-100) containing a protease inhibitor cocktail and phosphatase inhibitor. After 20 min of incubation on ice, samples were centrifuged at 12,000 rpm for 20 min at 4 °C. An equal amount of protein was mixed with 4X Laemmli buffer and heated at 95°C for 5 min. Samples were resolved by sodium dodecyl sulfate polyacrylamide gel electrophoresis and subsequently transferred to a polyvinylidene fluoride membrane by electroblotting. Membranes were incubated overnight at 4 °C with primary antibodies. After washing five times with Tris-buffered saline containing Tween 20, membranes were incubated with HRP-conjugated secondary antibodies. Signals were detected with Immobilon^®^ Crescendo Western HRP Substrate (Millipore).

### 2.9. Molecular Docking and Binding Site Prediction

For molecular docking analysis, the crystal structure of the Epo-EpoR complex (PDB ID: 1EER) [24] was obtained from the Protein Data Bank. Briefly, after downloading the receptor in .pdb format, the unused portion was removed, and the polar hydrogen group was added and given charge using AutoDockTools 1.5.6. Next, the optimized ligand structure was evaluated at the B3LYP/6-31 + G(d,p) level of theory using the Gaussian09 program and changing the format from .log to .pdb with Open Babel 2.4.1. Subsequently, the receptor and ligand were saved in pdbqt format using AutoDockTools 1.5.6. Finally, the ligand was docked with the Epo-EpoR complex using Autodock Vina 1.1.2 software from The Scripps Research Institute [25]. Discovery Studio Visualizer 19.1.0 was used to visualize docking results.

The binding site of the Epo-EpoR complex was obtained by a GHECOM web server using the mathematical morphology of pockets on protein surface [26].

### 2.10. Statistical Analysis

Statistical significance was determined by one-way ANOVA followed by post hoc (Bonferroni) multiple comparisons between treatment groups using Prism version 8.0 for Windows (GraphPad Software). A two-tailed unpaired Student’s *t-*test was used to compare the means of two groups. *p* < 0.05 was considered statistically significant.

## 3. Results

### 3.1. C. comosa Extract Enhanced Recovery from Anemia

To investigate the potential of *C. comosa* to alleviate anemia symptoms, severe anemia was induced in C57BL/6J mice by exposure to sub-lethal total body irradiation to deplete bone marrow cells. In Figure 1, numbers of RBCs in control animals were 10.27 ± 0.15 × 10^6^/mm^3^ before the induction of anemia and were progressively decreased with time, reaching the lowest level during days 14 to 18 (3.38 ± 0.25 × 10^6^/mm^3^ and 3.34 ± 0.25 × 10^6^/mm^3^ at day-14, and day-18, respectively). After day 18, RBC counts gradually increased with time. Treatment of the anemic mice with *C. comosa* extract at dosages of 50 and 100 mg/kg BW/day significantly increased RBC count (Figure 1A), hematocrit (Figure 1B), and Hb concentration (Figure 1C) from day 18 onward, but reticulocyte levels were decreased from day 26 (Figure 1D) of treatment compared with the vehicle control group. However, the extract slightly affected numbers of white blood cells and platelets (Appendix A). These results suggest that *C. comosa* extract improved RBC count and Hb content, subsequently ameliorating anemia symptoms in a mouse model of anemia.

### 3.2. ASPP 049- and E2-Enhanced Epo-Stimulated Murine Erythropoiesis

To explore the principle active component of *C. comosa* involved in enhancement of Epo-stimulated murine erythropoiesis, we investigated the effect of the major compound isolated from *C. comosa* (the diarylheptanoid, ASPP 049) on erythropoiesis using Lin^−^ murine HSCs. These cells, which were isolated from the bone marrow of C57BL/6J mice, were treated with ASPP 049 or E2 in a methylcellulose medium for 7 days. We found that in combination with a suboptimal concentration of Epo (0.05 U/mL), 0.1, 1, and 10 µM ASPP 049 or 10 and 100 nM E2 significantly increased numbers of BFU-E colonies (Figure 2B), whereas 50 µM ASPP 049 had no significant effect. In Figure 2C, 1 µM ASPP 049 and 10 nM E2 both significantly increased cumulative murine erythroid cell numbers in the presence of 0.1 U/mL Epo on days 3, 5, and 7 after treatment. These results demonstrated that both ASPP 049 and E2 promoted murine erythroid progenitor cell (BFU-E) numbers and erythroid cell proliferation in the presence of Epo.

### 3.3. ASPP 049 and E2 Enhanced the Effect of Epo to Promote Human Erythropoiesis

The effects of ASPP 049 on human erythropoietic cells derived from HSCs were next investigated using a CFC assay. As shown in Figure 3A, in combination with a suboptimal concentration of Epo (0.05 U/mL), 1 and 10 µM ASPP 049 significantly increased the number of BFU-E colonies formed by human CD34^+^ cells. In the absence of Epo, ASPP 049 (0.1, 1, and 10 µM) failed to induce the formation of BFU-E colonies (Figure 3A). Consistent with the observed proliferative effects, a combination of 0.1 U/mL Epo with 1 µM ASPP 049 or 10 nM E2 significantly increased numbers of cumulative erythroid cells on day 11 (Figure 3B). Therefore, 1 µM ASPP 049 and 0.1 U/mL Epo were chosen for subsequent experiments. The effect of ASPP 049 on erythropoiesis was further confirmed using cell surface markers of immature RBCs (CD71 and GPA). Combining Epo with 1 µM ASPP 049 or 10 nM E2 significantly increased expression of CD71 and GPA on days 10 and 14 of treatment (Figure 3C). By day 14, ASPP 049 and E2 had increased expression of these cell surface markers from 62.6% to 73.4% and 69.2%, respectively (Appendix A). Immunohistochemical staining of cytospin preparations of HSC-derived erythroblast cultures confirmed the effect of ASPP 049 in improving erythroid cell differentiation. On day 14 of culture, numbers of RBCs were greater in cultures treated with a combination of 0.1 U/mL Epo and 1 µM ASPP 049 compared with 0.1 U/mL Epo alone (Appendix A). For all experiments, 1 U/mL Epo was used as a positive control. Collectively, these results suggest that ASPP 049 and E2 promoted Epo-stimulated erythropoiesis in human HSC-derived erythroblasts.

### 3.4. ASPP 049 Acted through Erα Activation to Induce Human Erythropoiesis

ERα has previously been implicated in HSC division and erythropoiesis [10,11]. To determine whether ERα participates in the regulation of ASPP 049-induced erythropoiesis, CD34^+^ HSCs were treated with ASPP 049 or E2 in the presence of the ER antagonist ICI 182,780 as indicated. As shown in Figure 4A, on day 11, both ASPP 049 and E2 significantly increased the cumulative erythroid cell number, but these effects were decreased in the presence of ICI 182,780. Moreover, the induction of erythroid precursor cells (BFU-E colonies) mediated by ASPP 049 and E2 was abolished by co-treatment with ICI 182,780 (Figure 4B). However, with a higher dosage of Epo (1 U/mL), ICI 182,780 did not significantly reduce the enhancing effect of ASPP 049 or E2. The enhancing effect of ASPP 049 and E2 on the number of immature (CD71^+^GPA^+^) RBCs was slightly decreased by ICI 182,780 (Figure 4C). Collectively, these results suggest that ERα is required for ASPP 049 or E2 to enhance Epo-induced erythropoiesis in human HSC-derived erythroblasts.

### 3.5. ASPP 049 Itself Is Unable to Induce Erythropoiesis of Erythroid Precursor Cells

Erythroid precursor (CD36^+^) cells are the targets to activate human RBC production by erythropoietic stimulating agents [27]. To assess whether ASPP 049 can induce erythropoiesis, the CD36^+^ cells were treated with 1 µM ASPP 049 or 10 nM E2 in the presence or absence of 0.1 or 1 U/mL Epo for 4 days. As shown in Figure 5A, ASPP 049 nor E2 could induce the proliferation of erythroid precursor cells. Consistent with immature RBC populations, ASPP 049, and E2 both failed to increase numbers of CD71^+^GPA^+^ cells in the presence of a suboptimal concentration of Epo (0.1 U/mL); 1 U/mL Epo was used as a positive control (Figure 5B). Moreover, treatment with the ER-antagonist ICI 182,780 did not affect cumulative erythroid precursor cell number (Appendix A). Together, these results indicate that erythroid precursor cells are not a target for ASPP 049-induced erythropoiesis.

### 3.6. ASPP 049 Potentiated the Effects of Epo Innactivating ERK, AKT, and STAT5 Phosphorylation

Epo-EpoR signaling is a major pathway in erythroid differentiation [4]. After Epo binds to EpoR, the cell surface form of EpoR is internalized and degraded [28,29], subsequently inducing signaling cascades including ERK, AKT, and STAT5 pathways [4]. Therefore, we investigated whether activation of erythropoiesis by ASPP 049 is mediated through the EpoR signaling pathway. UT7/Epo cells were used to determine the effect of ASPP 049 with Western blotting. We found that 10 µM ASPP 049 and 0.1 U/mL Epo markedly reduced protein expression of the cell surface form of EpoR (upper band). Moreover, ASPP 049 progressively increased phosphorylation of AKT(Ser473) from 10 to 120 min, as well as ERK(Thr202/Tyr204) at 30 min after treatment. However, treatment with a physiological dosage of Epo, AKT(Ser473), and ERK(Thr202/Tyr204) phosphorylation appeared to strongly affect the expression of AKT-ser473 and ERK-Thr202/Tyr204 phosphorylated protein, as clearly detected at early point (10 min) compared with ASPP 049. Interestingly, treatment with a combination of Epo and ASPP 049 effectively induced phosphorylation of AKT(Ser473) and ERK(Thr202/Tyr204), indicating an additive effect of these two compounds (Figure 6A). As expected, phosphorylation of STAT5 was observed in ASPP 049- and Epo-treated UT7/Epo cells and was delayed (observed at 60 min) in a co-treatment condition (Figure 6B). These results were further confirmed to occur in a concentration-dependent treatment for 30 min, and we found that both ASPP 049 and Epo could reduce expression of the cell surface form of EpoR (Figure 6C) and increase phosphorylation of AKT(Ser473) and ERK(Thr202/Tyr204), as shown in Figure 6D. This information suggests that ASPP 049 is, at least in part, involved in the EpoR signaling pathway.

### 3.7. ASPP 049-Mediated Epo-Epor Complex in Inducing ERK and AKT Phosphorylation

To evaluate whether the Epo-enhancing effect of ASPP 049 was associated with EpoR-mediated signaling pathways, the binding affinity of ASPP 049 to EpoR was determined using molecular docking analysis. As ASPP 049 can modulate erythroid cell production only in the presence of Epo, a model of the Epo-EpoR complex was employed in our docking design. We found that ASPP 049 binds to this complex with a binding affinity equal to −6.2 kcal/mol (Figure 7A and Appendix A). This binding location was supported by GHECOM binding site prediction, as shown in the black circle in Appendix A (colors represent possible areas for ligand binding sites). The two-dimensional interaction profile of ASPP 049 and the Epo-EpoR complex is shown in Figure 7B. ASPP 049 could form interactions with Arg21, Leu18, and Phe208. The aromatic ring was stabilized by Glu25, Leu27, and Phe208, while the remaining residues (shown in light green) contributed to ASPP 049 binding via van der Waals interactions. These results indicated that ASPP 049 can bind to the Epo-EpoR complex and initiate EpoR signaling. To further prove whether the EpoR is required for the Epo-enhancing effect of ASPP 049, we knocked down the EpoR in UT7/Epo cells using EpoR siRNA. The siRNA decreased EpoR protein levels in the UT7/Epo cells by ~50% (Figure 7C,D) compared to the control. We then investigated the effects of ASPP 049 and Epo on signaling cascade including AKT and ERK phosphorylation in the EpoR-knockdown UT7/Epo cells using Western blotting. We found that neither ASPP 049 nor suboptimal Epo induced the phosphorylation of AKT(Ser473) and ERK(Thr202/Tyr204) in the EpoR-knockdown cells (Figure 7E,F). These effects remained intact in the siControl cells. Collectively, our data clearly suggest that ASPP 049 in cooperation with a suboptimal concentration of Epo activates the EpoR signaling cascades, leading to enhanced erythropoiesis.

## 4. Discussion

The present study demonstrated a novel role of the phytoestrogen diarylheptanoid, ASPP 049, isolated from *C. comosa*, in enhancing Epo-induced erythropoiesis. In a mouse model of anemia, *C. comosa* extract improved Hb content and RBC counts and ameliorated anemic symptoms. ASPP 049, the major compound isolated from *C. comosa*, was found to be the principal active component in exerting these effects. ASPP 049 enhanced Epo-induced erythropoiesis of murine Lin^−^ and human HSCs; however, its effect required the presence of Epo. The molecular mechanism by which ASPP 049 stimulated erythropoiesis was mediated through ERα and EpoR signaling cascades, i.e., AKT, ERK, and STAT5 pathways. In addition, the effect of ASPP 049 was observed only during the early stage of erythropoiesis in erythroid progenitors, the effect was absent in precursor (CD36^+^) cells. This finding has a great impact on inducing the generation of erythroid cells, especially correcting an early defect in erythropoiesis. Moreover, its biological activity in combination with Epo may have therapeutic potential in anemia resulting from different causes.

ASPP 049 exhibits estrogenic activity [17] and induces osteoblast differentiation through Wnt/β-catenin and MAPK signaling pathways [18,19]. Indeed, these two pathways have been demonstrated to promote erythropoiesis both in vivo and ex vivo [30,31]. In the present study, the effects of a diarylheptanoid ASPP 049 and the underlying mechanism were evaluated on murine- and human-models of erythropoiesis. ASPP 049 and E2 could induce BFU-E colony formation and HSC proliferation of Lin^−^ cells derived from mouse bone marrow. This result confirmed that the effect of *C. comosa* extract in accelerating recovery of anemic-induced mice involves the primary compound isolated from this extract, ASPP 049.

In human HSCs, both E2 and ASPP 049 synergistically worked with a suboptimal concentration of Epo to promote erythropoiesis, as observed by increased human BFU-E colony formation and erythroblast proliferation and differentiation. Although many factors are involved in control of processes for RBC formation, the Epo-EpoR signaling pathway is a major mediator [4]. Therefore, targeting this key pathway is commonly used for anemia treatment. Suboptimal production of erythropoietin in the kidney is one of the common causes of anemia in aging, in addition to bone marrow failure [32]. Currently, recombinant Epo is routinely used to treat patients with anemia caused by chronic kidney disease and myelosuppressive cancer chemotherapy [33]. However, treatment with high dosages of Epo is ineffective for some hematological diseases and may promote tumor progression [34,35]. Therefore, decreasing the concentration of Epo while maintaining effective treatment is ideal for medicating patients with anemia. Our findings indicate that Epo concentrations in our system were decreased to a level approximately 10 times lower than that of the normal condition (0.05–0.1 vs. 1 U/mL, respectively). However, the addition of 1 µM ASPP 049 together with 0.1 U/mL Epo clearly increased the immature erythrocyte population. Our results suggest an alternative treatment using a low concentration of Epo while maintaining therapeutic effectiveness, which would limit any adverse effect associated with conventional Epo-related therapy. As there are several causes contribute to anemia in aging, ASPP 049, which induces proliferation of HSCs and possesses anti-inflammation, may have potential for treating anemia in aging.

The estrogen receptor (ER) is known to be involved in erythropoiesis [36]. Our results indicate that both E2 and a phytoestrogen diarylheptanoid, ASPP 049, play a role in the regulation of RBC production through ERs. In the present study, HSCs treated with ICI 182,780 (an ER antagonist) exhibited clearly decreased HSC-derived erythroblast numbers and BFU-E colony formation (Figure 4). Based on studies from two independent groups [10,11], activation of ER signaling by a selective agonist of ERα (PPT) but not ER-β (diarylpropionitrile) significantly improved erythropoiesis, suggesting the involvement of ERα during erythropoiesis. Indeed, our previous study reported that ASPP 049 also acted on ERα for the activation of uterotrophic and estrogenic osteogenesis [17,18]. Taken together, our results suggest that ASPP 049 mediated ERα for activation of erythropoiesis in erythroblast-derived human CD34^+^ HSCs.

Production of RBCs is mainly influenced by EpoR activation and subsequent stimulation of three major signaling pathways: STAT5, MAPK/ERK, and PI3K/AKT [37]. To determine whether ASPP 049 plays a role in the regulation of these pathways in erythroid cells, we treated Epo-sensitive UT-7 cells with ASPP 049, Epo, or combination treatment. We observed that treatment with ASPP 049 or Epo clearly enhanced phosphorylation of STAT5, MAPK/ERK, and PI3K/AKT in both a time- and concentration-dependent manner, whereas a combination of ASPP 049 with suboptimal Epo prolonged activation of these pathways. These results corroborate well with a previous report showing that iron potentiated the action of Epo in erythroid cells through transferrin receptor functions [38]. Consistently, it has been demonstrated that iron supplementation decreases the requirement for Epo to support erythropoietic differentiation and improves anemia in patients with chronic disease [39]. In the present study, molecular docking analysis revealed that the susceptibility of ASPP 049 to bind to the Epo-EpoR complex, the initiating activator of STAT5, MAPK/ERK, and PI3K/AKT pathways. Thus, it is likely that ASPP 049 exerts its action similar to that of polymeric immunoglobulin A1 or iron on Epo-EpoR-induced erythropoiesis. Moreover, we observed that activation of these pathways correlated to reduced expression of the cell surface form of EpoR. Consistent with our findings, previous studies have shown that control of erythroid cell proliferation was mediated by rapid degradation of EpoR through lysosomal and proteasomal pathways [28,29]. However, the intermolecular link between EpoR-mediated pathways induced by ASPP 049 and EpoR degradation requires further investigation.

Inhibition and induction of cytochrome P450 (CYP) enzymes, which result in clinically significant drug–drug interactions, are commonly found in herbal remedies [40]. It is often a limitation of using herbal medicine. In the present study, although the *C. comosa* extract and the diarylheptanoid ASPP 049 have therapeutic potential with curative outcomes in anemia, their induction of some CYP enzymes may limit their clinical use. To minimize clinical risks before long-term application, they are required to assess these particular issues of herb–drug interactions. Another limitation is that the stimulation of STAT5, MAPK/ERK, and PI3K/AKT pathways, which are activated by ASPP 049, is also observed in the proliferative tissues including leukemia [41]. In addition, the estrogenic activity of the ASPP 049 may be able to affect other ERα-expressing target organs in the body. Long-term treatment of this compound therefore needs to be warranted. It is worth noting that ASPP 049 possesses a weak estrogenic activity compared with E2 [17], and treatment-related adverse effects of the compounds in both animal models and cellular levels have not been observed. To determine whether *C. comosa* and ASPP 049 are clinically significant, a study in humans warrants further application.

## 5. Conclusions

In conclusion, the diarylheptanoid ASPP 049 enhances the effects of Epo and displays estrogenic activity via ERα-mediated activation of STAT5, MAPK/ERK, and PI3K/AKT signaling pathways. These actions are conserved in both in vitro human erythroid cell-derived HSCs and an in vivo mouse model of anemia. Therefore, *C. comosa* could be used as an innovative therapeutic strategy in combination with physiological concentrations or lower Epo for the treatment of patients with anemia.

## Figures and Tables

**Figure 1 biomedicines-10-01427-f001:**
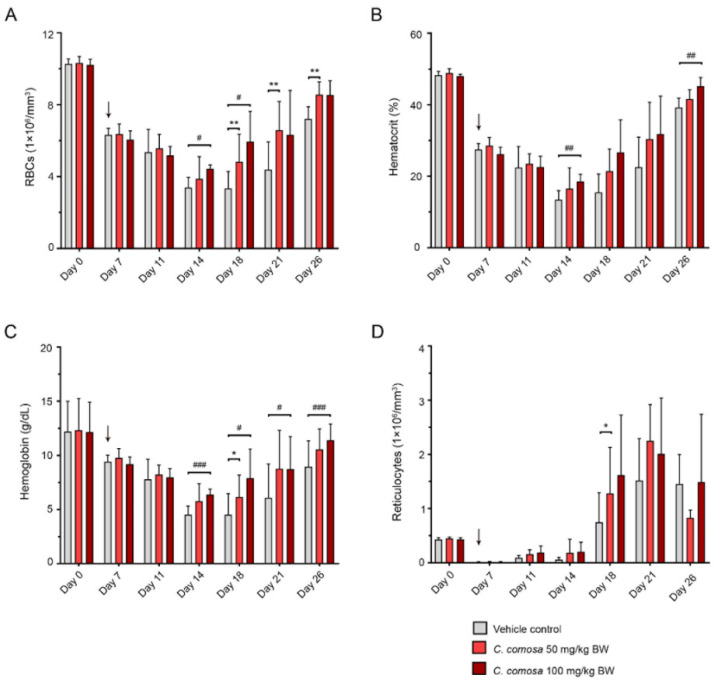
*Curcuma comosa* accelerated recovery from anemia. Erythropoietic response of mice to sublethal irradiation-induced anemia: (**A**) RBC counts, (**B**) hematocrit level, (**C**) hemoglobin level, and (**D**) reticulocyte counts at days 7, 11, 14, 18, 21, and 26. Irradiated mice were intraperitoneally injected with vehicle control or *C. comosa* (50 or 100 mg/kg BW) each day. The blood sample was collected from the retro-orbital plexus, and hematological parameters were monitored with an MS9 analyzer. All data are expressed as mean ± SEM (*n* = 9). * *p* < 0.05 and ** *p* < 0.01, ASPP 049 compared with vehicle control treatments (ANOVA); ^#^ *p* < 0.05, ^##^ *p* < 0.01, and ^###^ *p* < 0.001, E2 compared with vehicle control treatment (ANOVA). BW, body weight; RBC, red blood cell. Black arrow indicates the induction of anemia.

**Figure 2 biomedicines-10-01427-f002:**
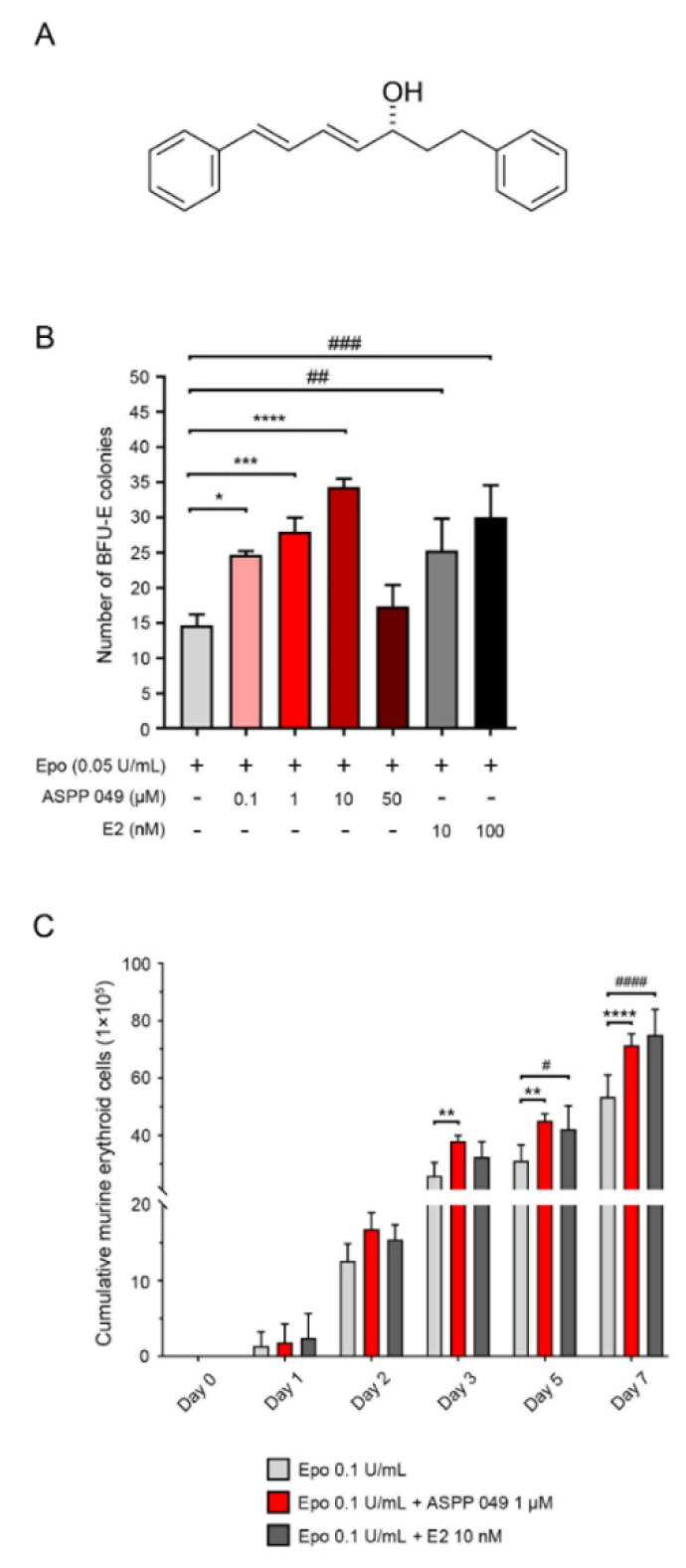
ASPP 049- and E2-induced murine erythroid progenitor cell formation. (**A**) Chemical structure of (3*R*)-1,7-diphenyl-(4*E*,6*E*)-4,6-heptadien-3-ol (ASPP 049), a diarylheptanoid isolated from *Curcuma comosa***.** (**B**) Concentration-dependent induction of mouse erythroid cells by ASPP 049 and E2. Mouse Lin^−^ bone marrow cells plated in MethoCult M3236 were treated with indicated concentrations of ASPP 049 or E2 in the presence of 0.05 U/mL Epo. At day 7, numbers of BFU-Es were analyzed according to the manufacturer’s recommendations (Stem Cell Technologies). (**C**) ASPP 049 increased proliferation of mouse Lin^−^ cells. Lin^−^ cells derived from mouse bone marrow were treated with 0.1 U/mL Epo, 0.1 U/mL Epo + 1 µM ASPP 049, or 0.1 U/mL Epo + 10 nM E2 for the indicated times. Proliferation of mouse Lin^−^ cells was observed by trypan blue exclusion assay. All data are expressed as mean ± SEM (*n* = 3). * *p* < 0.05 and ** *p* < 0.01, *** *p* < 0.001, **** *p* < 0.0001, ASPP 049 compared with vehicle control treatments (ANOVA); ^#^ *p* < 0.05, ^##^ *p* < 0.01, ^###^ *p* < 0.001, and ^####^ *p* < 0.0001, E2 compared with vehicle treatments (ANOVA). BFU-Es, burst-forming unit-erythroid cells; E2, 17β-estradiol; Epo, erythropoietin; HSCs, hematopoietic stem cells; Lin^−^, lineage negative.

**Figure 3 biomedicines-10-01427-f003:**
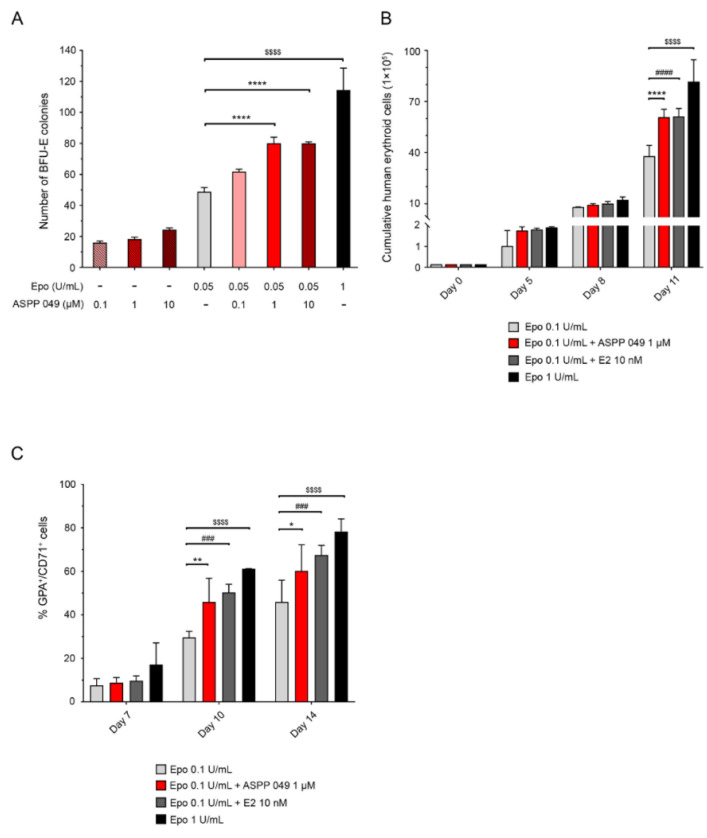
ASPP 049 and E2 increased erythropoiesis of human HSC-derived erythroblasts. (**A**) Human CD34^+^ HSCs plated in MethoCult H4236 were treated with various concentrations of ASPP 049 in the presence of 0.05 U/mL of Epo; 1 U/mL Epo was used as a positive control. Numbers of BFU-Es were scored on day 14 according to the manufacturer’s recommendations. (**B**) ASPP 049 and E2 increased human erythroid cell survival. CD34^+^ HSCs plated in a human erythroid culture medium were treated with 1 µM ASPP 049 and 10 nM E2 in the presence of 0.1 U/mL of Epo for the indicated times; 1 U/mL Epo was used as a positive control. HSC-derived erythroid cell survival was observed by trypan blue exclusion assay. (**C**) ASPP 049 and E2 increased human immature erythrocyte populations. CD34^+^ HSCs plated in a human erythroid culture medium were treated with 1 µM ASPP 049 and 10 nM E2 in the presence of 0.1 U/mL of Epo for the indicated times; 1 U/mL Epo was used as a positive control. Expression of immature erythrocyte markers (GPA^+^CD71^+^) was detected by flow cytometry analysis. All data are expressed as mean ± SEM (*n* = 3). * *p* < 0.05 and ** *p* < 0.01, **** *p* < 0.0001. ASPP 049 compared with vehicle control treatments (ANOVA); ^###^ *p* < 0.001 and ^####^*p* < 0.0001, E2 compared with vehicle treatments (ANOVA); ^$$$$^ *p* < 0.0001, 1 U/mL compared with vehicle control treatment (ANOVA). BFU-Es, burst-forming unit-erythroid cells; E2, 17β-estradiol; Epo, erythropoietin; HSCs, hematopoietic stem cells.

**Figure 4 biomedicines-10-01427-f004:**
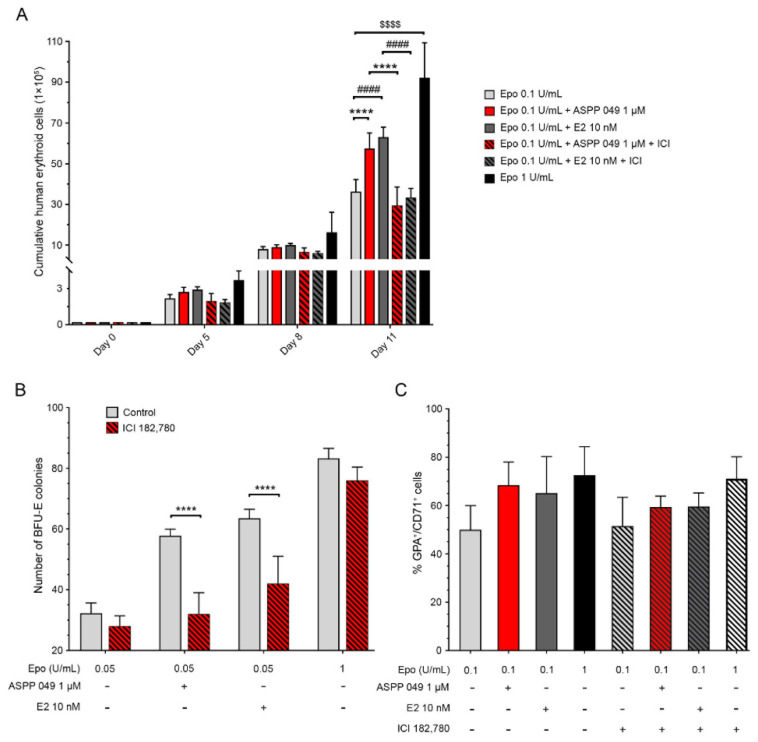
Induction of erythropoiesis by ASPP 049 and E2 requires estrogen receptors. (**A**) As indicated, CD34^+^ HSCs plated in a human erythroid culture medium were treated with 1 µM ASPP 049 and 10 nM E2 in the presence of 0.1 U/mL Epo or 1 µM ICI 182,780 (an estrogen receptor antagonist) for 5, 8, or 11 days; 1 U/mL Epo was used as a positive control. Survival of human erythroid-derived HSCs was observed by trypan blue exclusion assay. (**B**) As indicated, CD34^+^ HSCs plated in MethoCult H4236 were treated with 1 µM ASPP 049 and 10 nM E2 in the presence of 0.05 U/mL Epo or 1 µM ICI 182,780; 1 U/mL Epo was used as a positive control. Numbers of BFU-Es were scored on day 14 according to the manufacturer’s recommendations. (**C**) CD34^+^ HSCs were plated in human erythroid culture medium and treated with 1 µM ASPP 049 and 10 nM E2 in the presence of 0.1 U/mL of Epo or 1 µM ICI 182,780, as indicated. Expression of immature erythrocyte markers (GPA^+^CD71^+^) was detected by flow cytometry analysis. All data are expressed as mean ± SEM (*n* = 3). **** *p* < 0.0001, ASPP 049 compared with vehicle control treatments (ANOVA); ^####^ *p* < 0.0001, E2 compared with vehicle treatments (ANOVA); ^$$$$^ *p* < 0.001, 1 U/mL compared with vehicle control treatments (ANOVA). BFU-Es, burst-forming unit-erythroid cells; E2, 17β-estradiol; Epo, erythropoietin; HSCs, hematopoietic stem cells.

**Figure 5 biomedicines-10-01427-f005:**
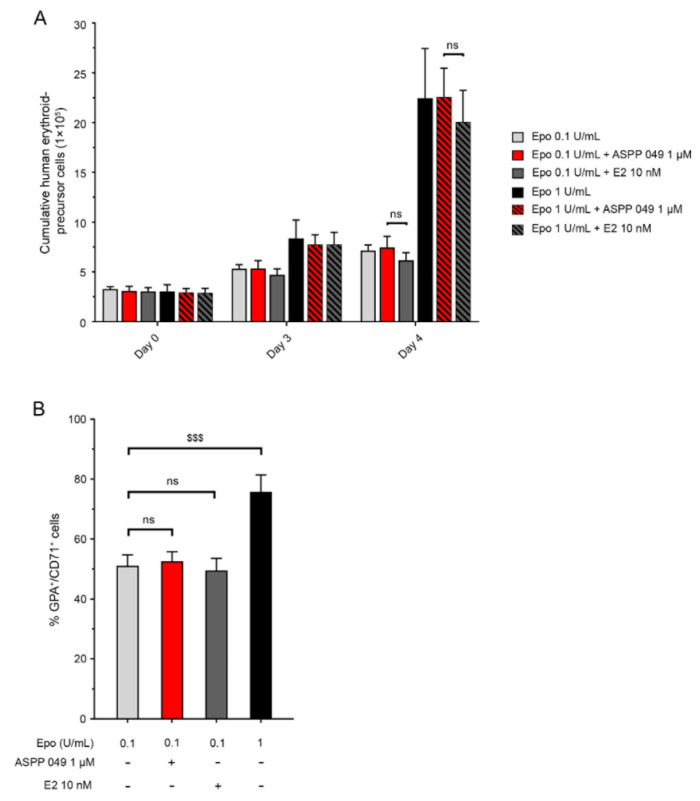
ASPP 049 could not stimulate erythropoiesis of CD36^+^ erythroid precursor cells. CD36^+^ cells sorted from CD34^+^ HSCs cultured in erythroid medium were treated with 1 µM ASPP 049 and 10 nM E2 in the presence of 0.1 or 1 U/mL of Epo. Proliferation of CD36^+^ cells was observed by (**A**) trypan blue exclusion assay and expression of immature erythrocyte markers (GPA^+^CD71^+^) was detected by (**B**) flow cytometry analysis. All data are expressed as mean ± SEM (*n* = 3). E2, 17β-estradiol; Epo, erythropoietin; HSCs, hematopoietic stem cells; ns, non-significant. ^$$$^ *p* < 0.001.

**Figure 6 biomedicines-10-01427-f006:**
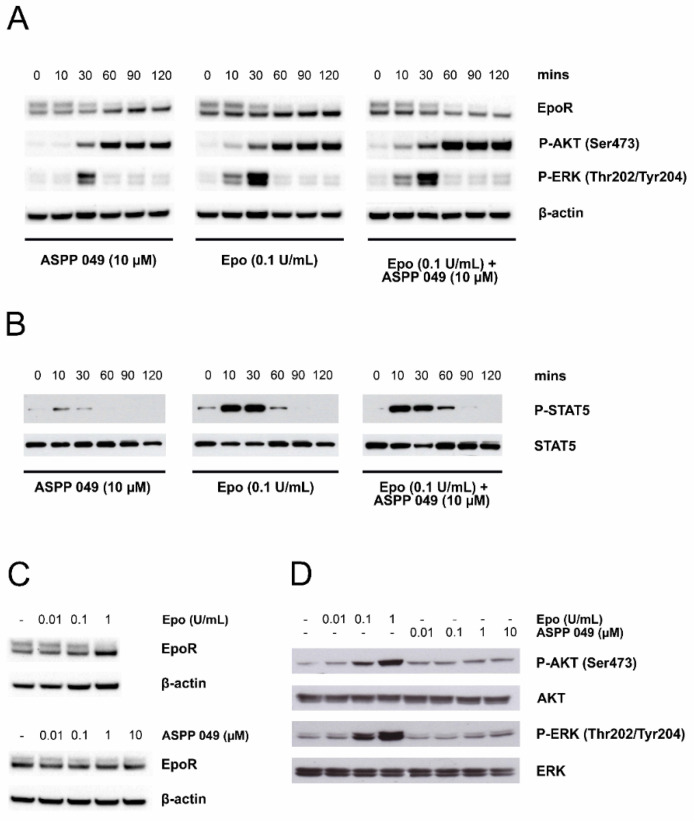
Stimulation of erythropoiesis by ASPP 049 involves EpoR-dependent activation of STAT5, MAPK/ERK, and PI3K/AKT pathways. (**A**) Representative blot showing time-dependent effects of ASPP 049, Epo, and ASPP 049 + Epo on protein expression of EpoR, phospho-AKT, and phospho-ERK. Serum-starved UT7/Epo cells were treated with 10 µM ASPP 049, 0.1 U/mL Epo, or 10 µM ASPP 049 + 0.1 U/mL Epo for different times (10–120 min) and harvested for immunoblotting with anti-EpoR, anti-phospho-AKT (Ser473), and anti-phospho-ERK (Thr202/Try204) antibodies. (**B**) Representative blot showing effects of ASPP 049 on phospho-STAT5 protein expression. Serum-starved UT7/Epo cells were treated with 10 µM ASPP 049, 0.1 U/mL Epo, or a combination for different times (10–120 min) and harvested for immunoblotting with anti-phospho-STAT5. (**C**,**D**) Representative blot showing concentration-dependent effects of ASPP 049 on EpoR, phospho-AKT, and phospho-ERK protein expression. Serum-starved UT7/Epo cells were treated with Epo (0.01, 0.1, or 1 U/mL) or ASPP 049 (0.01, 0.1, 1, and 10 µM) for 30 min and harvested for immunoblotting with anti-EpoR, anti-phospho-AKT (Ser473), and anti-phospho-ERK (Thr202/Try204). All data are representative of three independent experiments. AKT, protein kinase B; Epo, erythropoietin; EpoR, erythropoietin receptor; ERK, extracellular signal-regulated kinase; MAPK, mitogen-activated protein kinase; PI3K, phosphoinositide three kinases.

**Figure 7 biomedicines-10-01427-f007:**
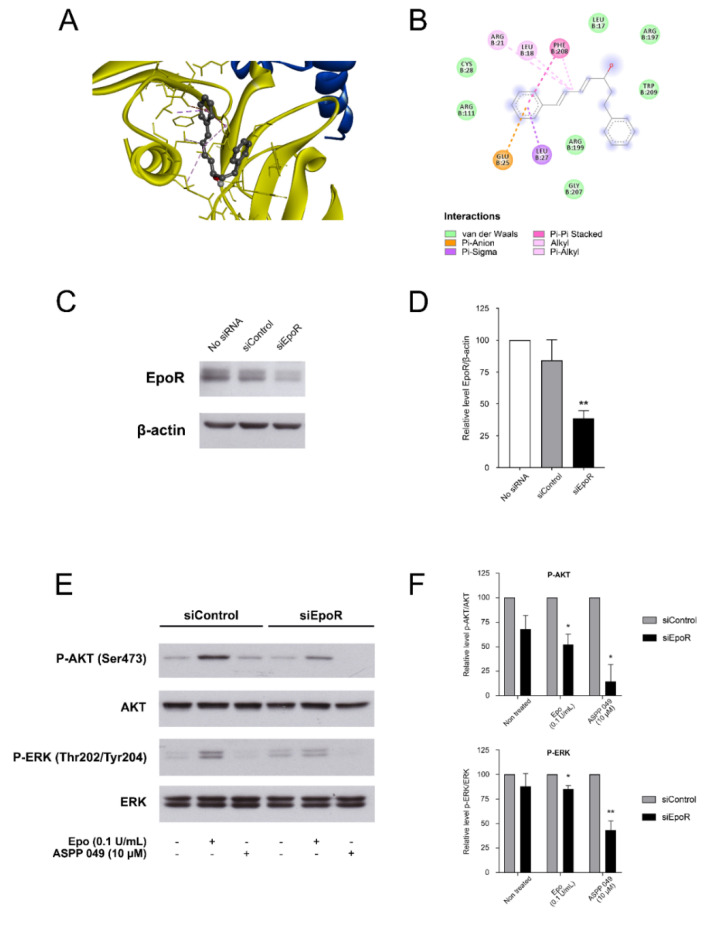
ASPP 049-mediated Epo-EpoR complex in inducing ERK and AKT phosphorylation. (**A**) Lowest energy position of ASPP 049 within the Epo-EpoR complex, crystal structure represented in gold. (**B**) Two-dimensional interaction profile of ASPP 049 at the binding site of the Epo-EpoR complex. (**C**) Representative blot showing the expression of EpoR in UT7/Epo cells (No SiRNA), or the cells transfected with non-targeting siRNA (siControl) or siEpoR for 24 h. β-actin was used as a loading control. (**D**) Graph showing the quantification of EpoR normalized to β-actin (*n* = 3). (**E**) Representative blot showing the effect of ASPP 049 and Epo on phospho-AKT and phospho-ERK protein expression in the presence of siEpoR. Serum-starved UT7/Epo cells were treated with 0.1 U/mL Epo or 10 µM ASPP 049 for 30 min and harvested for immunoblotting with anti-EpoR, anti-phospho-AKT (Ser473), and anti-phospho-ERK (Thr202/Try204). All data are representative of three independent experiments. (**F**) Graph showing the quantification of phospho-AKT (Ser473) and phospho-ERK (Thr202/Try204) normalized to total AKT and total ERK, respectively. All data are representative of three independent experiments. * *p* < 0.05 and ** *p* < 0.01, ASPP 049 or Epo compared with control treatments (Student’s *t*-test).

## Data Availability

Not applicable.

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
