# Peer review of "Enhancing Erythropoiesis by a Phytoestrogen Diarylheptanoid from Curcuma comosa"

_biomedicines, 2022, doi:10.3390/biomedicines10061427_

Round 1
Reviewer 1 Report
Review molecules-1738355-R1
This manuscript’s title" Enhancing erythropoiesis by a phytoestrogen diarylheptanoid from Curcuma comosa" As shown in the manuscript, the authors wanted to e investigate the potential effects of a phytoestrogen diarylheptanoid (ASPP 049) isolated from Curcuma comosa on the specific interaction with Epo-EpoR in a hematopoietic system. However, the authors must be noticed some comments below.
Comments to Authors
1. In this manuscript, the authors used a phytoestrogen diarylheptanoid (ASPP 049) isolated from Curcuma comosa as the agent to interact with Epo-EpoR. There are several diarylheptanoids were isolated from Curcuma comosa. However, the authors did not describe any information about this compound. We also can not find any information about ASPP 049. Suggest the authors must supplement the scientific name, chemical structure, and structure identification information and explain this compound well.
2. Always, phytoestrogens mainly belong to the coumestans, prenylflavonoids, and isoflavones. However, diarylheptanoid does not belong to them. Only two papers used this term (Including one paper from the authors), so it is easy to confuse the reader. Why do the authors name the compound as phytoestrogen diarylheptanoid? Suggest the authors explain them well.
3. This manuscript described diarylheptanoid (ASPP 049) induced the phosphorylation of AKT, ERK, and STAT5. However, the other common diarylheptanoid (curcumin) which was the same isolated from Curcuma comosa always showed reduced phosphorylation of AKT, ERK, and STAT5[1-3]. Why did the similar diarylheptanoids compound isolated from Curcuma comosa but had opposing effects on the phosphorylation of AKT, ERK, and STAT5t? What can the readers believe? Suggest the authors explain them well?
4. It is well known that activation of the JAK/STAT, PI3K/AKT, and ERK signaling Is involved in the mediated promotion of inflammation and stimulated invasion and migration of cancer cells[3, 4]. Diarylheptanoid ASPP 049 enhances the activation of STAT5, MAPK/ERK, and PI3K/AKT 493 signaling pathways. Is ASPP 049 suited to improving erythropoiesis in anemia patients without no harm? Suggest the authors explain them well.
5. Reduced tissue oxygenation and increased production of erythropoietin by hypoxia-inducible factor 1 (HIF-1) were two major roles in enhancing erythropoiesis[5, 6]. In this study, why the authors did not do any studies with them? Did this experiment, ASPP 049 could be induced a hypoxia state or induced the produced excess HIF? Suggest the authors explain them well.
6. In this manuscript, the author only detect the expression of EpoR by western blot but did not assay any EpoR-related signaling pathway or hypoxia state or induced the produced excess HIF experiments. It lacks evidence to prove any mechanism. What is the major theory of enhancing erythropoiesis the authors want to explore? Suggest the authors explain them well.
7. In Figure 7A, ASPP 049 binds to EpoR complex with a binding affinity equal to -6.2 kcal/mol it seems lower docking ability, seem can not support those related findings. In Figures 7C and D, the data does not relate to ASPP 049. Suggest the authors explain them well.
Reviewer 2 Report
The study entitled – “Enhancing erythropoiesis by a phytoestrogen diarylheptanoid 2 from Curcuma comosa” by Bhukai K is an interesting work. There are certain concerns which needs to be addressed before it can be published.
The comments are attached below;
1 . The authors have only specified about the beneficial aspect of phytoestrogen diarylheptanoid (ASPP 049 isolated from Curcuma comosa in this work. It is very important to know what toxic effects this agent will have on xenobiotic metabolism. It is mandatory to measure AST, and ALT levels in the plasma and also if possible major CYP enzyme expressions, (like CYP3A4, CYP2D6) in the liver tissue following C. comosa extract 50 or 100 mg/kg of body weight (BW)] dose.
2 . The understand the in vivo effects of ASPP-049 it is ideal to measure systemic inflammatory marker expressions for C-reactive protein, IL-6 and TNF α.
3 . In the discussion section, please include a sub section with limitations of the study. Please explain in detail in a small paragraph that what could be limitations for using this compound. It should be based on your study and also available literature in database, so that readers are aware of the complications and pitfalls associated with the use of ASPP-049 from Curcuma comosa. This will definitely strengthen the manuscript.
Round 2
Reviewer 1 Report
No more extra comments.
This manuscript is a resubmission of an earlier submission. The following is a list of the peer review reports and author responses from that submission.